# A Low-Cost Automatic Vehicle Identification Sensor for Traffic Networks Analysis

**DOI:** 10.3390/s20195589

**Published:** 2020-09-29

**Authors:** Fernando Álvarez-Bazo, Santos Sánchez-Cambronero, David Vallejo, Carlos Glez-Morcillo, Ana Rivas, Inmaculada Gallego

**Affiliations:** 1Department of Civil and Building Engineering, University of Castilla-La Mancha, 13071 Ciudad Real, Spain; fernando.alvarezbazo@uclm.es (F.Á.-B.); ana.rivas@uclm.es (A.R.); inmaculada.gallego@uclm.es (I.G.); 2Department of Technologies and Information Systems, University of Castilla-La Mancha, 13071 Ciudad Real, Spain; david.vallejo@uclm.es (D.V.); Carlos.Gonzalez@uclm.es (C.G.-M.)

**Keywords:** plate scanning, low-cost sensor, sensor location problem, traffic flow estimation

## Abstract

In recent years, different techniques to address the problem of observability in traffic networks have been proposed in multiple research projects, being the technique based on the installation of automatic vehicle identification sensors (AVI), one of the most successful in terms of theoretical results, but complex in terms of its practical application to real studies. Indeed, a very limited number of studies consider the possibility of installing a series of non-definitive plate scanning sensors in the elements of a network, which allow technicians to obtain a better conclusions when they deal with traffic network analysis such as urbans mobility plans that involve the estimation of traffic flows for different scenarios. With these antecedents, the contributions of this paper are (1) an architecture to deploy low-cost sensors network able to be temporarily installed on the city streets as an alternative of rubber hoses commonly used in the elaboration of urban mobility plans; (2) a design of the low-cost, low energy sensor itself, and (3) a sensor location model able to establish the best set of links of a network given both the study objectives and of the sensor needs of installation. A case of study with the installation of as set of proposed devices is presented, to demonstrate its viability.

## 1. Introduction

### 1.1. The Purpose and Significance of This Paper

The monitoring of traffic in urban networks, whatever their complexity, is a problem that has been tackled for decades. The aim of this monitoring depends on the case and can involve managing the daily traffic flow to perform urban mobility plans. Regarding the techniques and tools to identify and quantify the vehicles on the network, traditional manual recording has been displaced by more sophisticated techniques due to their economy, and also to collect the traffic information with enough performance and quality. Basically, the emerging techniques consist of a sensor or device able to collect a type of information through its interaction with a vehicle or the infrastructure. Therefore, the sensors used for traffic analysis can be classified in different categories according to their physical characteristics, type of collected information, and position with respect to the network among others. In particular, [1] differs between in-vehicle and in-road sensors. The first are those that allow increasing the performance of the driving and the connectivity of the vehicles with their environment. In this, the concepts of communication between vehicle and the vehicular sensing networks (VSN) are called to be important in the improvement of the quality and operability of transportation systems (see [2,3]). The second are those installed in the transportation network and allows the monitoring of the performance of the system and, according to the extracted information, diagnose the problems, improve the resilient and operational functioning, and inform the users helping them to make better choices. In this paper we mainly focus in this last. Based on the works of [4,5], in-road sensors for traffic network analysis are classified in two main groups according to the characteristics of the data collected (see Figure 1):
Sensors for punctual data collect traffic information at a single point of the road, and can be designed to obtain information for each single vehicle (e.g., vehicle presence, speed, or type), or for the vehicles in a defined time interval (e.g., vehicles count, average speed, vehicles occupancy, etc.). In addition, these sensors can be
○“Passive sensors” do not require any active information provided from a vehicle, i.e., they collect the information when a vehicle is passing in front of the sensor. In particular:
▪“Passive fixed sensors” have a fixed position on the network. This group includes inductive loop detectors, magnetic detectors, pressure detectors, piezoelectric sensors, microwave radars, among others. These sensors are used to manage the traffic and can also be used to elaborate traffic mobility plans using only the already installed fixed sensors if the available budget is limited.▪“Passive portable sensors” have a fixed position on the network, but they are installed for a defined-short period of time. This group includes counters made with rubber hoses or manual counters that are used for example to elaborate traffic mobility plans completing the information provided by fixed sensors.○“Active sensors” require active information from the vehicle to be univocally identified. In fact, these sensors can be included under the term “automatic vehicle identification” (AVI). As well as the passive sensors they can be fixed or portable:
▪“Active fixed sensors” have a fixed position on the network. This group includes automatic number plate recognition (ANPR) sensors, Bluetooth sniffer of bar-coded tags. Despite these sensors being designed for other purposes far from the traffic network analysis, recent researches have begun to use the data collected by these sensors to estimate traffic flows.▪“Active portable sensors” have to be designed to be installed for a very short period of time to take info from vehicles. As far as we know this kind of sensor has a very limited use for traffic management and more for Police controls, such as ANPR sensors, both those that are temporarily installed on the road and those installed on vehicles (although this latter could be also considered as in-vehicle sensors).“Sensors for section data” are those that collect information in different sections on the network providing the number of vehicles traveling from different points in the network, travel times between these points, entrances and exits between reidentification devices, etc. This group includes mainly sensors for license plate recognition, but other approaches that allow vehicle re-identification to match measurements at two (or more) data collection sites that belong to the same vehicle. In general, all these sensors are installed fixed in the network.

The data collected by the sensors can be used for multiple purposes but, since this paper is focused on the topic of traffic flow estimation, only those used as inputs for these models are going to be analyzed. These sensors have to satisfy two objectives: accuracy and coverage [6] and, due to their ease installation and capability of data collection, passive sensors (e.g., fixed loop detectors or portable rubber hoses) have been widely used in mobility studies in large urban areas.

As exposed above, sensors as rubber hoses count the number of vehicles that pass over it, obtaining the needed traffic counts used by traditional methods to estimate origin–destination (O–D), route and link flows on a network. The quality of the results of this estimate may be enough for some cases, but when the technicians or the authorities look for a better degree of observability (or even full observability) of traffic flows to achieve a high quality of estimation, the traffic count data has been proved to be not sufficient. For this, it is expected that these sensors are going to be gradually replaced by new active sensors (as ANPR) that, taking advantage of the available technology and the added value provided by the data, allows the development of models to better estimate the non-observed flows.

### 1.2. State of the Art of Sensors for ANPR

The automatic number plate recognition (ANPR) system is based on image processing techniques to identify vehicles by their number plates, mainly in real time (for automatic control of traffic rules). In [7] or also in [8] a review is made regarding the most significant research work conducted in this area in recent years.

The general process of automatic number plate recognition can be summarized in several well-defined steps [9,10]. Each step involves a different set of algorithms and/or considerations:Vehicle image capture: This step has a critical impact on the subsequent steps since the final result is highly dependent on the quality of the captured images. The task of correctly capturing images of moving vehicles in real time is complex and subject to many variables of the environment, such as lighting, vehicle speed, and angle of capture.Number plate detection: This step focuses on the detection of the area in which a number plate is expected to be located. Images are stored in digital computers as matrices, each number representing the light intensity of an individual pixel. Different techniques and algorithms give different definitions of a number plate. For example, in the case of edge detection algorithms, the definition could be that a number plate is a “rectangular area with an increased density of vertical and horizontal edges”. This high occurrence of edges is normally caused by the border plates, as well as the limits between the characters and the background of the plate.Character segmentation: Once the plate region has been detected, it is necessary to divide it into pieces, each one containing a different character. This is, along with the plate detection phase, one of the most important steps of ANPR, as all subsequent phases depend on it. Another similarity with the plate detection process is that there is a wide range of techniques available, ranging from the analysis of the horizontal projection of a plate, to more sophisticated approaches such as the use of neural networks.Character recognition: The last step in the ANPR process consists in recognizing each of the characters that have been previously segmented. In other words, the goal of this step consists in identifying and converting image text into editable text. A number of techniques, such as artificial neural networks, template matching or optical character recognition, are commonly employed to address this challenge. Since character recognition takes place after character segmentation, the recognizer system should deal with ambiguous, noisy or distorted characters obtained from the previous step.

Once the data is collected by the sensors, it has to be properly processed to be used for a great amount of traffic analysis. In particular, focusing on the scope of traffic flow analysis, the data allows to
Develop models where the observable flows are directly related with the routes followed by the vehicles [11,12,13]. Since both link flows and O–D flows can be directly derived from route flows, these models are a powerful tool for traffic flow estimation.Extract a great amount of information compared with traffic counts which in turn permits developing a model with more flow equations for the same number of variables [14].Obtain the full observability of the traffic flows if the budget is sufficient to buy the needed number of sensors [15,16].Being combined with other sources of data to improve the results [17].Measure other variables as travel times in traffic networks if the location of sensors is adequate [4,18].

An extra step to complement the aforementioned steps is the error recovery that may occur when recognizing plate numbers. This problem is a very important issue to deal with when plate scanning data is used for traffic flow estimation, which some authors have been faced using different approaches [16,19,20].

However, the increasing development of these ANPR systems faces some problems such as: they are fixed sensors and they incur a high cost in terms of hardware [21] (about $20,000 per camera) and installation and maintenance (about $4000 per camera). This makes necessary to develop new architectural approaches that allow these types of services to be deployed on a larger scale to face transportation problems such as urban mobility plans. It is worth noting the survey collected in [22], which analyzes the sensors to monitor traffic from the point of view of various criteria, including cost. In this study, it is highlighted that the new sensors tend to be of reduced dimensions, of low energy consumption and that, with a certain number of them, it is possible to design and configure a sophisticated wireless sensor network (WSN) that can cover multiple observations in a certain region [23,24].

Regarding existing software libraries and tools focused on automatic plate recognition, “OpenALPR” (2.5.103) [25] stands out. This open source library, written in C++, is able to analyze images and video streams to automatically identify license plates. The generated output is a text representation that comprises the set of characters associated with each one of the identified plate numbers. The hardware required to run OpenALPR depends on the number of frames per second that the system must handle. From a general point of view, a resolution of 5–10 fps is required for low-speed contexts (under 40 km/h), 10–15 fps for medium speed contexts (40–72.5 km/h), and 15–30 fps for high speed contexts (over 72.5 mph). The library requires significant computing power, with the use of several multi-core processors at 3 GHz to process images at 480 p in low-speed contexts. From the point of view of the success rate, OpenALPR represents the software library with the best results on the market (more than 99% success in a first estimation [26]).

“Plate Recognizer” (1.3.8) [27] offers cloud-based license plate recognition services for projects with special needs such as diffuse, low-light, or low-resolution imaging. The cloud processing pricing plan offers different configurations per processing volume. There are also other specific purpose platforms for automatic license plate identification in the market, such as “SD-Toolkit” (1.2.50) [28], “Anyline” (24) [29], or the framework “Eocortex” (3.1.39) [30].

In recent years, conventional ANPR systems are strengthening their services through the use of AI techniques [31]. “Intelli-Vision” (San Jose, CA, USA) [32], the company that offers intelligent image analysis services using AI and deep learning techniques, has specific license plate recognition services that can be integrated, via an existing SDK, in Intel processors or provided as a web service in the cloud. The Canadian company “Genetec” (Montreal, Quebec, Canada) [33] announced, at the end of 2019 an ANPR camera that includes an Intel chip designed to feed neural networks improving the identification of license plates at high speed or in bad weather conditions.

Finally, it is very important to keep in mind if the ANPR systems can respect the users’ privacy rights in the entire process in which the vehicle data is collected according to the different locations all along the network [34]. All this means that, when designing a type of sensor that can be implemented in an architecture that serves to monitor the traffic network, the cost criteria for manufacturing and installation, operability and resilience, and information processing must been taken into account.

### 1.3. Contributions of This Paper

It is being seen how the sensors based on the capture of vehicle images constitute an efficient traffic monitoring system for its features. However, there is still a challenge in terms of manufacturing and installation costs, since well-designed equipment and materials are required in terms of performance and functionality to face different network conditions [19,34]. This is a very important challenge because the large number of papers published by researchers in recent years (see [35] or [4] for a good review), stated that in order to achieve good traffic flow estimation results, a large number of sensors has to be installed. Even when trying to minimize this number the model developed in [36] proposed to install 200 ID-sensors to obtain the full observability of a real size city with 2526 links. Depending on the case of study, this can be an unaffordable cost. In addition, the sensor location models have to be designed to take into account the particular characteristics of installation of the type of sensor to be used.

Therefore, all the context exposed in this section motivates the preparation of this original paper, whose main contributions are as follows:A novel architecture to deploy low-cost sensor networks able to automatically recognize plate numbers, which can be temporarily installed on city streets as an alternative to rubber hoses commonly used in the elaboration of urban mobility plans.A design of a low-cost, low-energy sensor composed of a number of hardware components that provides flexibility to conduct urban mobility experiments and minimize the impact on maintenance, installation, and operability.A methodology to locate the sensors able to establish the best set of links of a network given both the study objectives and of the sensor needs of installation. This model integrates the estimation of traffic flows from the data obtained by the proposed sensors and also establishes the best set of links to locate them taking into account the special characteristics of its installation. Furthermore, using the proposed methodology, we have proved that the expected quality of the traffic flow estimation results are very similar if the sensor can be located in any link compared with avoiding links with certain problems to install the sensor.

The rest of the paper is organized as follows: in Section 2, the proposed low-cost sensor and its associated system for traffic networks analysis are deeply described. In Section 3 the proposed system is applied in a pilot project in Ciudad Real (Spain). Finally, some conclusions are provided in Section 4.

## 2. The Proposed Low-Cost ANPR System for Traffic Networks Analysis

This section deals with the description of the proposed system which is composed of three elements: (1) the proposed architecture to deploy the sensor networks, (2) the devised low-cost sensor prototype, and (3) the adopted method to decide the best set of links where the sensors have to be installed.

### 2.1. Architecture to Deploy Low-Cost Sensor Networks

#### 2.1.1. General Overview

Figure 2 shows the multi-layer architecture designed to deploy low-cost sensor networks for automatic license plate detection. The use of a multi-layer approach ensures the scalability of the architecture, as it is possible to carry out modifications in each of the layers without affecting the rest. In particular, the architecture is composed of three layers:The perceptual layer, which integrates the self-contained sensors responsible for image capture. Each of these sensors integrates a low-power processing device and a set of low-cost devices that carry out the image capture. The used camera enables different configurations depending on the characteristics of the urban environment in which the traffic analysis is conducted.The smart management layer, which provides the necessary functionality for the definition and execution of traffic analysis experiments. This layer integrates the functional modules responsible for the configuration of experiments, the automatic detection of license plates, from the images provided by the sensors of the perceptual layer, and the permanent storage of information in the system database.The online monitoring layer, which allows the visualization, through a web browser, of the evolution of an experiment as it is carried out. Thanks to this layer, it is possible to query the state of the different sensors of the perceptual layer, through interactions via the smart management layer.

#### 2.1.2. Perceptual Layer

The perceptual layer is the lowest-level layer of the proposed architecture. It contains the set of basic low-cost processing sensors that will be deployed in the physical environment to perform the image capture. In this layer, these sensors are not aware of the existence of the rest of the sensors. In other words, each sensor is independent of the others and its responsibility is limited to taking pictures at a certain physical point, sending them to the upper-level layer, and, periodically, notifying they are working properly. Since the sensor design represents a major component of the proposed architecture, a detailed description of its main characteristics is done in Section 2.2.

#### 2.1.3. Smart Management Layer

The smart management layer provides the functionality needed to (i) facilitate the deployment of low-cost sensor networks and the execution of experiments, (ii) process the images captured by the perceptual layer sensors, performing automatic license plate detection, and (iii) persistently store all the information associated with an experiment for further forensic analysis.

This layer of the architecture follows a Platform-as-a-Service (PaaS) model, i.e., using the infrastructure deployed in the cloud, which provides the computational needs of the traffic analysis system. This approach makes it possible to offer a scalable solution that responds to the demands of the automatic license plate detection system, avoiding the complexity that would be introduced by deploying our own servers to provide functional support.

Particularly, “Google App Engine” [37] has been used as a functional support for the system’s server, using the Python language to develop the different components of the system and the Flask web application framework to handle web requests. The information retrieval with respect to the sensors of the perceptual layer is materialized through web requests, so that these can ask for the initial configuration of a sensor, or send information, as the so-called “control packages”, as the state of a traffic analysis experiment evolves.

In this context, the control package concept stands out as the basic unit of information to be handled by the smart management layer. The control package is composed of the following fields:Client ID. Unique ID of the sensor that sends the packet within the sensor network.Timestamp. Temporal mark associated with the time when the sensor captured the image.Latency. Latency regarding the previously sent control packet. Its value is 0.0 for the first control packet.Plates. List of candidate plate numbers, together with their respective confidence values, detected in the image captured by the sensor.Image. Binary serialization of the image captured by the sensor.

There are three different modules in this layer, which are detailed next:Experiment definition module: This module is responsible for managing high-level information linked to a traffic analysis experiment. This information includes the start/end times of the experiment and the configuration of the parameters that guide the operation of the perceptual layer sensors. This configuration is retrieved by each of these sensors through a web request when they start their activity, so that it is possible to adjust it without modifying the status of the sensors each time it is necessary.Processing module: This module provides the functionality needed to effectively perform automatic license plate detection. Thus, the input of this module is a set of images, in which vehicles can potentially appear, and the output is the set of detected plates, together with the degree of confidence associated with those detections. In the current version of the system, the commercial, web version OpenALPR library is used [26]. This module is responsible for attending the image analysis requests made by the sensors of the perceptual layer. Both the images themselves and the license plate detections associated with them are reported to the database management and storage module.Database management and storage module: This module allows the permanent storage of all the information associated with a traffic analysis and automatic license plate detection experiment. At a functional level, this module offers a forensic analysis service of all the information generated as a result of the execution of an experiment.

It is important to note that the processing module offers two modes of operation: (i) online and (ii) offline. In the online mode, the processing module carries out an online analysis of the images obtained from the perceptual layer, parallelizing the requests received by them to provide results in an adequate time. In contrast, the offline mode of operation is designed to analyze large sets of images associated with the past execution of a traffic analysis experiment.

#### 2.1.4. Online Monitoring Layer

The general objective of this layer is to facilitate the monitoring, in real time, of the evolution of a traffic analysis experiment. In order to facilitate the use of the system and avoid the installation of software by the user, the interaction through this layer is made by means of a web browser. From a high level point of view, the online monitoring layer offers the following functionality:Overview of the system status: Through a grid view, the user can visualize a subset of sensors in real time. This view is designed to provide a high level visual perspective of the sensors deployed in a traffic analysis experiment. It is possible to configure the number of components of the grid.Analysis of the state of a sensor: This view makes it possible to know the status, in real time, of one of the previously deployed sensors (see Figure 3). In addition to visualizing the last image captured by the sensor, it is possible to obtain global statistics of the obtained data and the generated information if an online analysis is performed.

In both cases, the information represented in this layer, through a web browser, is obtained by making queries to the layer of the immediately lower level, that is, the smart management layer. The latter, in turn, will obtain the information from the perceptual layer, where the sensors deployed in the physical scenario are located.

#### 2.1.5. Systematic Requirements

This subsection presents a well-defined set of systematic requirements provided by the devised architecture, considering the practical deployment of low-cost sensor networks for ANPR. These requirements are as follows:Scalability, defined as the architectural capacity and mechanisms provided to integrate new components.Availability, defined as the system robustness, the detection of failures, and the consequences generated as a result of these failures.Evolvability, defined as the system response when making software or hardware modifications.Integration, defined as the capacity of the architecture to integrate new devices.Security, defined as the ability to provide mechanisms devised to deal with inadequate or unauthorized uses of the deployed sensor networks.Manageability, defined as the capacity to interact between the personnel responsible to conduct the experiments and the software system.

Regarding scalability, the architecture proposed in this work provides support (i) when new low-cost sensors need to be integrated and (ii) when new physical locations need to be monitored. The integration of new sensors is carried out in the perceptual layer. Thus, this systematic requirement is guaranteed thanks to the existing independence between sensors. As mentioned before, each sensor is responsible for a single physical point. Similarly, when a new physical location needs to be added, then it is only necessary to deploy a new sensor which, in turn, will send information to the upper-level layer and will notify whether it is working properly. This is why integration is also guaranteed in terms of adding new devices when they are required. In other words, this requirement is strongly related to scalability of the devised architecture.

The systematic requirement named availability has been achieved thanks to the adopted cloud-based approach, since it is easy to incorporate multiple layers of license plate analysis so that processing errors are identified. Although the currently deployed system only uses OpenALPR, the architecture easily allows the incorporation of other license plate identification platforms that minimize potential errors. On the other hand, all processing sensors run the same software on the same hardware. Replacing a sensor implies changing its identifier and the server address that are specified in the configuration file stored in the memory stick. In other words, replacing a faulty sensor is a simple and straightforward task. This decision is related to the systematic requirement evolvability.

With respect to security, multiple methods have been considered to protect the information exchanged between the different components of the architecture, ensuring its integrity. Particularly, the extension hypertext transfer protocol secure (HTTPS) has been used to guarantee a secure communication so that the information is encrypted using secure sockets layer (SSL). Finally, regarding manageability, the devised architecture aims at facilitating the deployment process of sensor networks. In fact, there is a component, named experiment module definition, which has been specifically designed to address this systematic requirement. As previously stated in Section 2.1.3, it is possible to set up experiments and adjust the configuration of the sensors in a centralized way, without having to individually modify the internal parameters of every single sensor.

### 2.2. Low Cost Sensor Prototype Description

#### 2.2.1. Production Cost

From a hardware point of view, each low-cost sensor (€62.27) is composed of the following components (see Figure 4):Raspberry Pi Zero W: €11.97Raspberry Pi Camera Module V2.1: €23.63Power bank PowerAdd Slim2 5000 mAh: €8.293D plastic box (PLA): €1.18MicroSD card U1 16 GB: €6.49Memory stick 128 MB: €1.97Tripod: €10.74

Raspberry Pi is a low-cost single board computer running open source software. The multiple versions of the board employ a Broadcom processor (ARM architecture) and a specific camera connector. Thanks to the use of this hardware, the versions of the Raspberry Pi OS (formerly called “Raspbian”), derived from the GNU/Linux distribution Debian, can be used. Thus, the development in a number of general-purpose programming languages is possible.

For the development of the sensor previously introduced, the version of the board called Pi Zero W has been used, which incorporates the Broadcom BCM2835 microprocessor. This has a single-core processor running at 1 GHz, 512 MB of RAM, a VidoCore IV graphics card, and a MicroSD card as a storage device. Based on the Pi Zero model, this version offers Wi-Fi connectivity, which allows online monitoring. In the conducted tests, the connectivity with the cloud has been done by using 3G/4G connection sensors, using the existing institutional Wi-Fi network of the University of Castilla-La Mancha whenever possible.

The 8 megapixel Raspberry Pi Camera V2.1 features Sony’s IMX219RQ image sensor with high sensitivity to harsh outdoor lighting conditions, with fixed pattern noise and smear reduction. The connection is made using the camera’s serial interface port directly to the CSI-2 bus via a 15-pin flat cable. The camera automatically performs black level, white balance, and band filter calibrations, as well as automatic luminance detection (for changing conditions) of 50 Hz in hardware. In the configuration of each sensor, the resolution with which each photograph is taken can be specified, up to the maximum of 3280 × 2464 pixels.

The used Lithium battery holds a capacity of 5000 mAh, with an output of 5 V/2.1 A and a very small size and weight (100 × 33 × 31 mm, 195 g).

The installed operating system is based on Debian Buster, with kernel version 4.19. The installation image has a size of 432 MB which, once installed on the system partition, uses 1.7 GB. The current version of the prototype uses a UHS Speed Class 1(U1) microSD card, with a write speed of 10 MB/s required to record high-definition pictures in short intervals. Each 8 MP photograph may require around 4 MB in jpg format if stored at full resolution (depending on the scene complexity and lighting conditions).

In the conducted tests, each sensor made the captures with a resolution of 1024 × 720 pixels. Each image occupied an average of 412 KB, size that was reduced to 151 KB after the optimization process with capture sub-regions. The capture frequency was established to 1 image per second. This requires a disk storage of 1.4 GB for every hour of capture without optimization. Thus, with more than 14 GB available on the SD card for data, it is possible to store more than 8 h of images without optimization, and more than 24 h by defining capture sub-regions.

The 128 MB Flash Drive is used to store the processing sensor configuration parameters, such as the unique identifier of the processing sensor, the address of the web server associated with the intelligent experiment management layer, and the network configuration.

For the integration of all hardware components of the system, a basic prototype has been made using 3D printing, with a size of 103 × 78 × 35 mm, and a unit cost of 1.18 (59 g of PLA of 1.75 mm).

The cost of the designed sensor is similar to some of the low-cost sensors discussed in [22]. However, the offered functionality can be compared to commercial systems with a significantly greater cost. Plus, the devised architecture enhances the global functionality of the sensor networks deployed from the architecture, and this is a major improvement regarding existing work in the literature.

#### 2.2.2. Energy Cost

The energy cost of the system depends mainly on the use of the processor. In the deployed system, the most expensive computational stage is done in the cloud, so three working states can be defined in the sensors: the “idle” mode, in which the sensor is waiting for work orders, the “capture” mode, in which de sensor accesses the camera and saves the image in the local storage, and the “networking” mode, which optimizes the image with the defined sub-regions and sends them to the smart management layer.

Table 1 summarizes the power consumption between different versions of Raspberry Pi (all fice versions). The ZeroW version was chosen because it provides wireless connectivity (not available on Zero), and because of the very low power consumption it requires (0.6 Wh in idle mode, 1 Wh in capture mode, and 1.19 Wh in networking mode). In this way, a small 10 W solar panel could be enough to provide the energy required by the sensor.

#### 2.2.3. Maintenance, Installation, and Operability

The use of a general purpose processor, such as the Broadcom BCM2835, facilitates rapid prototyping, as well as the integration of existing software modules. In particular, the integration of the functionality offered to the smart management layer is done in a straightforward way thanks to this approach.

On the other hand, the impact of maintenance costs and the addition of new functionality is minimized by using a cloud-based approach where each sensor is configured through specific parameters. A unique identifier and server address are specified for each sensor. From the server, the sensor receives a JSON message with the parameters to be used in each analysis experiment. By using this configuration package per sensor, it is possible to adjust the specific capture configuration of each sensor in the network, based on its position, weather conditions, or lighting level at each time of day. For example, a sensor that may be better positioned to identify license plates will be able to take lower resolution captures (saving processing costs) than a sensor that is located further away from the traffic. Even the same sensor may need to make higher resolution captures in adverse weather situations, such as rain or fog.

The JSON message has the same format:
{
“begTime”: “2020-06-10T09:00:00”,“endTime”: “2020-06-10T11:00:00”,“resolution”: “1024x720”,“mode”: “manual”,“exposure_time”: 1000,“freq_capture”: 1000,“iso”: 320,“rectangle_p1”: [
280,262],“rectangle_p2”: [
1024,574]}

The fields begTime and endTime indicate the date and time of the start and the end of the capture session. Resolution indicates the capture resolution of the sensor with values supported by the hardware up to a maximum of 3280 × 2464 pixels. If the field mode is set to manual, it is possible to indicate the shutter speed or exposure time, which defines the amount of light that enters the camera sensor. The parameter exposure_time defines the fraction of a second (in the form 1/exposure seconds) that the light is allowed to pass through. The field freq_capture indicates the number of milliseconds that will pass between each capture. The field iso defines the sensitivity of the sensor to light (low values for captures with good light level). Finally, the fields that begin with the keyword rectangle allow us to define capture sub-regions within an image. The upper left and lower right corners define the valid capture rectangle within the image. The rest of the pixels are removed from the image, facilitating the transmission of the image through the network and avoiding storage and processing costs in regions where plate numbers will never appear (see Figure 5).

The use of parameters that are used to define sub-regions in the captured images, their size can be drastically reduced. Any 3G connection is more than enough to cover the bandwidth requirements of each processing sensor, without any loss of image quality. Even under more adverse transmission conditions (such as Enhanced Data rates for GSM Evolution (EDGE) or General Packet Radio Service (GPRS) coverage with maximum speeds between 114 and 384 Kbps), the frame could be stored using a higher level of JPG compression without significant loss of image quality (up to a level of 65 would be acceptable), and therefore without putting at risk the identification of the license plate (see Figure 6).

#### 2.2.4. Information Processing

The proposed architecture, and particularly the smart management layer that was previously discussed, improves the processing costs, offering results that can be in real time or with programmed offline execution. In this way, the use of the platform as a whole can even be shared between different sets of sensors, avoiding the unnecessary complexity of local management at the level of each sensor or group of sensors.

From the point of view of information processing, it is possible to minimize the information traffic between the image analysis system (in the cloud) and the processing sensors. As a way of example, Figure 7 shows how the sensors can make fewer requests by encoding multiple captures into one single image.

### 2.3. Methodology to Locate ANPR Sensors in a Traffic Network

Having described the sensors to be located and its operating system, the next step is to determine their best locations on the network. To do this, given (1) a reference demand and traffic flow conditions; (2) a traffic network, defined by a graph (N,A), where N is the set of nodes and A is the set of links; and (3) the budget of the project (i.e., a number of available sensors), the next aim is to obtain the locations that allow obtaining the best possible traffic flow estimation. Depending on the number of sensors to be located, we can achieve total or partial observability of the network according to the flow conditions and the number of routes modelled on it. The suitable locations for these sensors are determined from the use of two algorithms that integrate the previous three elements. In this section, these two algorithms are described.

#### 2.3.1. Algorithm 1: Traffic Network Modelling

The method used to build an appropriate network model, given a graph (N,A), for traffic analysis using plate-recognition based data is the one proposed in [13]. We assume that every node of the network can be the origin and the destination of trips, and therefore the classic zone-based D–D matrix has to be transformed into a node-based O–D matrix used as reference. This matrix is assigned to the network using a route enumeration model. Then, a route simplification algorithm is proposed based on transferring to adjacent nodes the generated or attracted (reference) demand of those nodes that generate or attract fewer trips than a given threshold. Figure 8 shows the operation of this first algorithm that involves the modeling of the network, and whose steps are described below.
**INPUT:** A traffic network (N,A) and its link parameters (links cost, links capacities, etc.); an out-of-date O–D matrix defined by traffic zones; capacities of links to attract and generate trips; the k parameter of the MNL assignment model; and the threshold flow ($F_{thres}$) to simplify the node-based O–D matrix and its corresponding routes.**OUTPUT:** A set Q of representative routes of the network and its corresponding route flow $f_{q}^{0}$, and a set of “real” data necessary to check the efficiency of the estimation.
STEP 1:Obtain the node-based O–D matrix: Given an O–D matrix by traffic zones in the network, and from some data on the attraction and trip generation capacities of the links that form it (see [13] for more details), it is possible to obtain an extended O–D matrix by nodes, defined as follows:(1)$$T_{ij}={\hat{T}}_{ZiZj}PA_{i}PG_{j}$$
where $T_{ij}$ is the number of trips from node i to node j; ${\hat{T}}_{ZiZj}$ is the number of trips from zone of node i to the zone of node j; $PA_{i}$ is the proportion of attracted trips at node i; and $PG_{j}$ is the proportion of generated trips at node j  which depends on its capacity to attract or to generate trips.STEP 2:Obtain the set R of reference routes: After defining the O–D matrix, an enumeration model, based on Yen’s k-shortest path algorithm [38], is used to define those k-shortest routes between nodes, which are then assigned to the network through a MNL Stochastic User assignment model. This model makes it possible to build an “exhaustive reference set of routes” R between nodes, with its respective route flows $f_{r}^{0}\;$,which will be operated by the algorithm, and whose size will vary according to the value adopted by the parameter k. Along with these reference data, other data considered as “real” will also be defined that will serve to check the effectiveness of the model in the flow estimation results obtained from the information collected by the sensors.STEP 3:Initialize the traffic network model simplification: The intention of this step is to adapt the modelled traffic network as close to the actual network as possible, simplifying those routes by O–D pairs whose attraction/generation trip flow is below a given threshold flow value. To do this, we initialize the set Q of modelled routes to the set R of reference routes.STEP 4:Evaluate the trip generations or attractions of the nodes: The algorithm evaluates the trips generated and/or attracted of each node of the network and compares them with the threshold flow value $F_{thres}$. If there is any node that holds this condition, go to Step 5, otherwise the algorithm ends and a simplified set of routes Q will be obtained, whose size will be a function of the value of the $F_{thres}$ flow considered.STEP 5:Transfer the demand: The node that meets with the condition in Step 4, would lose its generated/attracted demand, which would also imply that no route could begin or end from that node. Therefore, it will be necessary to transmit these flow routes to another node close enough (which could receive or emit demand) with an implicit route, whenever possible. If the demand transfer could not be carried out, the evaluated demand is lost and all the involved routes as well.STEP 6:Update the set of routes: Q**.** The set Q and its associated flows $f_{q}^{0}$ have to be updated with the deleted or updated routes. The O–D Matrix $T_{ij}$ must also be updated. Go to Step 4.

#### 2.3.2. Algorithm 2: The ANPR Sensor Location Model

After defining the traffic network, i.e., the set of routes and its associated reference flows, both are introduced in the location model so that from these, and with the particularities of the sensor to be used, this model allows us to obtain a set of links, SL, to locate a certain number of sensors to collect data able to obtain the best possible estimation of the remaining flows of the network. This can be a difficult combinatorial problem to solve, especially when it is required to locate sensors in large networks with a great number of existing routes (this justify the use the set of routes Q instead of set R). Next, we propose an iterative problem-solving process to find the best possible solution given a series of restrictions. Figure 9 shows the operation of this second algorithm, and whose steps are described below.
**INPUT:** A traffic network (N,A); Sets of routes R and Q, with their associated flows; the budget to install sensors B; an optional set of non-candidate scanned links NSL; and maximum number of iterations to be performed $iter^{max}$.**OUTPUT:** The set of scanned links $SL^{best}$ and the evolution of the RMARE value according to the performed iterations.
STEP 1:Solve the optimization problem: The following optimization problem has to be solved:(2)$$\;\underset{z_{a};y_{q}}{\max}M=\underset{q\in Q}{\sum }f_{q}^{0}y_{q}$$
subjected to
(3)$$\underset{a\in A}{\sum }P_{a}z_{a}\leq B\;\;\;\;\;\forall a\in A$$
(4)$$\underset{a\in A}{\sum }\delta_{a}^{q}z_{a}\geq y_{q}\;\;\;\;\;\forall q\in Q$$
(5)$$\underset{a\in A|\delta_{a}^{q}+\delta_{a}^{q1}=1}{\sum }z_{a}\geq y_{q}\;\;\;\;\;\forall \left(q,q1\right)\in Q^{2}|\;q>\;q1;\;\underset{a\in A}{\sum }\delta_{q}^{a}\;\delta_{q_{1}}^{a}>0$$
(6)$$z_{a}=0\;\;\;\;\;\forall a\in NSL$$
(7)$$\underset{a\in A}{\sum }S_{a}^{iter}z_{a}\leq \underset{a\in A}{\sum }S_{a}^{iter}\;\;\;\;\;\forall iter\in I$$The objective function (2) maximizes the distinguished route flow in terms of $f_{q}^{0}$; $y_{q}$ is a binary variable equal to 1 if a route can be distinguished from others and 0 otherwise. Constraint (3) satisfies the budget requirement, where $z_{a}$ is a binary variable that equals 1 if link a is scanned and 0 otherwise. This constraint guarantees that we will have a number of scanned links with a cost $P_{a}$ for link a that does not exceed the established limited budget B.Constraint (4) ensures that any distinguished route contains at least one scanned link. This constraint is indicated by the parameter $\delta_{a}^{q}$, which is the element of the incidence matrix. Constraint (5) is related to the previous constraint since it indicates the exclusivity of routes: a route q must be distinguished from the other routes in at least one scanned link a. If $\delta_{a}^{q}$ + $\delta_{a}^{q1}$ = 1, this means that the scanned link a only belongs to route q or route q1. If $\sum z_{a}\geq y_{q}$ and $y_{q}$ = 1, then at least one scanned link has this property; on the other hand, if $y_{q}$ = 0, then the constraint always holds.Constraint (6) is an optional constraint that allows a link to not be scanned if it belongs to a set of links not suitable for scanning NSL. This restriction will make the binary variable $z_{a}$ equal to 0, and therefore a sensor cannot be located on it. The intention of defining this constraint will be discussed with more detail in the next section.Finally, since this model is part of an iteration process (see [13] for more details), an additional constraint (7) is proposed, which allows us to obtain different solutions of SL sets for each iteration performed through the definition of $S_{a}^{iter}$, which is a matrix that grows with the number of iterations I, in which each row reflects the set SL resulting from each iteration carried out up to then by the model. Therefore, if an element of $S_{a}^{iter}$ is 1, means that link a was proposed to be scanned in the solution provided on iteration iter and 0 otherwise. Each iteration keeps the previous solutions and does not permit the process to repeat a solution in future iterations. That is, each iteration carried out by the algorithm is forced to search for a different solution $\;SL^{iter}$ with the same objective function (2).STEP 2:Simulate the sensor deployment and the “real” data sets: After obtaining the set $SL^{iter}$, the numerical simulation of it on the traffic network is carried out with the flow conditions given by an assumed “real” condition. One of the main features introduced in that algorithm is the possibility of working with a set of routes not fixed. Until now, the sensor location and flow estimation models have been formulated considering a set of existing fixed and non-changing routes. In the proposed model, each set SL may allow to obtain different observed set of combinations of scanned links (OSCSL) used by the vehicles (i.e., sets of links where vehicles have been registered), denoted by s. Since the modelled network and routes are not always the same as in reality, not all sets in OSCSL are compatibles with set of routes Q and hence new routes have to be added conforming a new global set C that encompasses the routes in Q and the new ones, with their associated flows $f_{C}^{0}$. To define these new sets from new routes discovered from this simulation, the algorithm looks for and assimilates their compatibility with those routes from set R that were eliminated in the simplification step of Algorithm 1 (see [13] for more detail). With this step, each set s of observed combinations of scanned links will provide the observed flow values ${\bar{w}}_{s}$ as the input data for the estimation model. In addition, apart from allowing to quantify the flow in routes from the scanned sets of links, these sensors behave as traffic counters in the link where they are installed, making it easier to quantify the flow in the link as well ${\bar{v}}_{a}$.STEP 3:Obtain the remaining traffic flows: In this step, a traffic flow estimation of the remaining flows is performed where route ($f_{c}$) and links ($v_{a}$) flows are obtained from reference flows ($f_{c}^{0}$) and the observed flows (${\bar{w}}_{s}$ and ${\bar{v}}_{a}$). We propose to use a Generalized Least Squares (GLS) optimization problem [14,15], as follows:(8)$$\underset{f_{c};\;v_{a}}{\min}Z\;=\underset{c\in C}{\sum }U_{c}^{-1}{\left(\frac{f_{c}-f_{c}^{0}}{f_{c}^{0}}\right)}^{2}+\underset{a\in SL}{\sum }Y_{a}^{-1}{\left(\frac{v_{a}-{\bar{v}}_{a}}{{\bar{v}}_{a}}\right)}^{2}$$
subjected to
(9)$${\bar{w}}_{s}=\underset{r}{\sum }\beta_{s}^{c}f_{c};\;\;\;\forall s\in S$$
(10)$$v_{a}=\underset{r}{\sum }\delta_{a}^{c}f_{c};\;\;\;\forall a\in A$$
(11)$$f_{c}\geq 0;\;\;\;\forall c\in C$$
(12)$$v_{a}\geq 0;\;\;\;\forall a\in A$$
where $U_{c}^{-1}$ and $Y_{a}^{-1}$ are the inverses of the variance–covariance matrices corresponding to the flow in route C and the observed flow in link a; ${\bar{w}}_{s}$ is the observed flow in each set OSCSL; $f_{c}\;$ is the estimated flow of routes in set C; $\beta_{s}^{c}$ and $\delta_{a}^{c}$ are the corresponding incidence matrices of relationship between observed link sets s and links a with routes.STEP 4:Check the quality of the solution. Once the flow estimation problem has been resolved, the quality of the solution in absolute terms, can be quantified as follows:(13)$$RMARE=\frac{1}{n}\underset{a\in A}{\sum }\frac{\left|v_{a}-v_{a}^{real}\right|}{v_{a}^{real}}$$
where RMARE is the root mean absolute value relative error; n is the number of links in the network; and $v_{a}$ and $v_{a}^{real}$ are the estimated flow and (assumed) real flow for link a. Such error is calculated over the link flows since the number of them remain constant regardless of the network simplification and the SL set used. Each value of RMARE indicates the quality of the estimation by using the set SL for the traffic network. As said above, due to the complexity of the problem, it has not a unique solution so we propose to evaluate a great amount of combined solutions in an iterative process. This iterative process, which is shown in Figure 9, is carried out since Step 1 a number of iterations equal to the maximum considered $iter^{max}$. For the solution found in the first iteration, the value of RMARE will be considered as the best, but in the following iteration, the algorithm could find another solution with lower value of and it will be considered as best. All the solution found and tested in each iteration are stored in $S_{a}^{iter}$ matrix, which grows in size during the performance. Finally, the best solution or set $SL^{best}$, for the established conditions, will be the one provided with the lower RMARE value.

## 3. The Application of the Proposed System in a Pilot Project

In this section, the proposed low-cost system for traffic network analysis is applied in a pilot project in a real network to demonstrate its viability and also to test the influence that some inputs of the Algorithm 1 (i.e., the network modelling) have in the results of the Algorithm 2 (i.e., the expected traffic flow estimation quality).

### 3.1. Description of the Project and Particularities about the Position of Sensors in the Streets

The network chosen to develop the pilot project was the traffic network of the University Campus of Ciudad Real (Spain), delimited in Figure 10, consisting of 75 nodes and 175 links. To consider the influence that the other districts have on this network, links connected to the contour were also modelled. With this, the set of links, its capacity and cost characteristics are established (these characteristics are available by requesting them to the corresponding author). The O–D matrix ${\hat{T}}_{ZiZj}$ was first defined, where each zone Z contains a certain set of nodes (see Table 2) resulting in a total of 15 zones as shown in Figure 10, while the reference O–D matrix by zones is shown in Table 3.

From a technical point of view, installing each traffic sensor in the streets of a city can be a complex task depending on the configuration and characteristics of the network and the elements that configure it. In the case of links or linear elements, the configuration of their infrastructure and the flow conditions, such as its distribution and intensity along a day, may condition the choice of one or another location, or even consider whether or not a link is a candidate for a sensor to be located. This section deals with the specific problems of installing the sensor in some links.

After a first test of the sensor in the streets of the project, we found a set of links that, due to their physical characteristics, may difficult the sensor to be installed. As shown in Figure 11a in violet, this set is formed by the links that make up the external corridor that connects the ends of the network, which is one of the main arterials of the city. In this type of links (see images 1 and 2 in Figure 11b), the vehicles can reach higher speeds and flow densities, which can make it difficult to capture the data because the license plate is not read correctly due to the occlusion of other vehicles as these links have two lanes per direction. Installing sensors in links where their characteristics make such a task difficult, may involve higher installation and/or operating costs, increasing the possibility that the data that they collect may have errors that may disturb the results of the analysis and the estimation of the remaining flows. The problems that wrong readings of plate license may involve on the flow estimation results have been studied in detail by [20]. Despite these links being very important since the greatest flow of vehicles takes place in them as they are one of the most important arterial corridors of the city. Therefore, the effects of not locating a sensor on them must be investigated.

With respect to the rest of links (see images 3 and 4 in Figure 11b), the results obtained from the sensor were very satisfactory resulting all of them suitable and hence being eligible. The flow conditions make feasible the correct identification of the vehicles regardless of its speed and flow density.

To sum up, the sensor location model described in Algorithm 2, has to consider the possibility of avoiding some links which, despite the fact that the greatest flow is concentrated by them, their characteristics make their installation difficult. This may have an impact on the results, since it seeks to obtain the best estimate of flows in the network with the best combination of scanned links. This observation is considered in the sensor location model with the inclusion of constraint (6), which has been described as a restriction that considers that for the arcs belonging to the NSL set, their binary variable is null, and therefore they are not suitable for having a sensor installed. Considering this topic can put a risk in obtaining better or worse estimation results, so an analysis is necessary to show that, by avoiding these links, the expected results of the traffic estimation can be similar. Next section below deals with a deeper analysis.

Finally, within the links that are suitable to be scanned, it is important to assess the different locations in them to obtain the correct reading of the license plates (see Figure 12). Here it is necessary to consider the orientation of the sensor with respect to the flow (i.e., recorded from the front or rear of the vehicle); the presence of fixed elements or obstacles present that make it difficult to identify the vehicle; the lighting among others.

### 3.2. Analysis of the Results

To obtain the traffic network model, we have applied the proposed Algorithm 1, where, in addition to the above described input data, the important values of k and $F_{thres}$ need to be defined. Therefore, with object to check the network simplification effects (Steps 4 to 6) on the estimation results (obtained with the Algorithm 2), it was decided to vary the value of the threshold flow $F_{thres}$, establishing values equal to 10, 15, 20, 25, and 30 trips per hour. Regarding the k parameter used in the enumeration model of Step 2, there are usually certain discrepancies between transportation analysts and engineers about its best value. High values are usually rare to find in the literature due to the high computational cost that it would entail, and also because the existence of more than 3–4 routes per O–D is very unlikely [39]. For this pilot project, it was considered to select reasonable values of k equal to 2 and 3, whose effects on results will be analyzed in the next sections. In Step 1, a 15 × 15 matrix by zones $\hat{T}$ shown in Table 2 was transformed into a matrix discretized by nodes T with size 75 × 75, resulting in a total of 608 O–D pairs. To obtain the set of existing routes assumed to be “real”, the node-based O–D matrix T is affected by a random uniform number (0.9–1.1), and the assignment was done using k=4 with to obtain the respective “real” link flows.

Regarding Algorithm 2, some of its inputs come from the outputs of Algorithm 1 (the traffic network and the routes). Due to the budget restrictions of the project (related to B parameter), an amount of 30 sensors was set to be used in the network. Therefore, for the different models studied, a fixed value B equal to 30 has been considered. Note that in relation to the number of links in the network, this quantity may be insufficient to obtain total observability, but it will be interesting to see to what degree of good estimation it is possible to obtain.

To sum up, in this section, three analysis of results are carried out:
An analysis of the effect on the estimation of flows is performed when considering different  k values for the definition of set R (Step 2 of Algorithm 1)**.** We have considered two values: 2 and 3.An analysis over the variation of the value of the threshold flow $F_{thres}$ that is used in the simplification algorithm is done (Steps 4 to 6 of Algorithm 1). Depending on the value of this threshold value, the degree of simplification of the network will be greater or lesser, affecting the number of considered routes in Q.An analysis to verify the effect of considering a certain set of links as not suitable to locate the scanning sensor (Equation (6) in Step 1 of Algorithm 2).

#### 3.2.1. Effect of Varying the k Parameter

Vary the k parameter means more or less number of routes in the modelled traffic network are considered, conforming part of the information with which the model must work. The consideration of such a parameter in this project has been through the use of a route enumeration algorithm, selecting values of 2 and 3 for the example presented. For this first analysis, it has been considered to analyze a not very simplified network scenario, considering a $F_{thres}$ equal to 10, i.e., all the nodes that attract or generate less than 10 trips lose its condition of origin and/or destination.

An important aspect that has been studied in this first analysis is related to the consideration of a set of links NSL∈A, where the cost and difficulties of installing a scanning sensor are greater than other links in the network. For the shake of brevity, we have decided to undertake a joint study of the k parameter influence and the effects of including some conflicting links in set NSL. A first scenario (Model A) where all the links have the same opportunity of install a sensor, which means that all the links have the same cost $P_{a}$ equal to 1. In the second scenario (Model B) a certain set of links (those corresponding to corridor shown in Figure 11a), are included in set NSL so a sensor cannot be installed in them.

Figure 13 shows the effects of these considerations on the results of the model. There are four well-differentiated lines in pairs, one assigning a k equal to 2 and another equal to 3. Each jump in the graph means that the location model has found a better set of scanned links SL that improves the solution in terms of error, and the horizontal sections mean that the model has not been able to find a better solution.

It is observed that considering a higher value of k, the results of the model are better in terms of the error in the estimation of traffic flows. We clearly see how a k = 3 obtains quite better results than considering a k = 2 due to the existence of a higher number of routes per each O–D pair. In particular, when considering k = 2, we are operating with a set R of 2074, as opposed to the 2943 routes considering k = 3. To define the set of “real” routes and their associated flows, a value of k = 4 has been considered, resulting in a set of 4274 routes in total.

The most interesting demonstration arises when Model A and Model B are compared. Despite considering a certain amount of links in set NSL, the results of both models reach almost the same RMARE value. We therefore see, in this particular case, how considering or not certain links to install the sensors does not produce a relevant difference in the estimation error to be obtained. In view of this, the following analysis will only consider Model B to avoid installation problems.

Table 4 shows the best SL sets obtained for each case after completing the iterative process. In it, the links that are common in both sets are marked in bold, seeing how a certain amount of them remains fixed, and the others are changing due to the modification of the location model through the constraint (6). This is clearly seen in Figure 14, where the optimal locations of the sensors are outlined in 30 of the links that make up the network. In this it is seen how a set of sensors, marked in blue, are located in NSL, i.e., when Model A was used. For Model B, it is seen how those sensors are moved to other links, now marked in orange. This change in location leads to an improvement in the estimation results, indicating that there would be no problem locating sensors in links in which, despite having a lower flow, there is a greater probability of obtaining data with lower mistakes.

#### 3.2.2. Effect of Network Simplification

When the value of $F_{thres}$ is small, the proposed methodology will do a smaller simplification of the network, and therefore it is expected to lead to a lower error in the estimation of traffic flows. As $F_{thres}$ increases, there will be a greater degree of simplification, and therefore greater error in the estimation. Figure 15 shows this effect all the cases modelled with k=3. It is observed that lower $F_{thres}$ values, and therefore less simplification, tend to smaller error values. In any case, depending on the $F_{thres}$ value, the graphs reached to a certain convergence after having performed multiple iterations with the proposed location model. For example, we see how the best solution is achieved with a minimum error difference when considering a $F_{thres}$ equal to 10 or 15 and for $F_{thres}$ equal to 20, 25, and 30.

The effects of variation in threshold flow are shown in Table 5. In it, a first column is defined for each evaluated case, and a second that collects the number of routes in the set R, which is the same for all of them since the same value of k = 3 was used; third column collects the number of routes set Q once set R has been simplified with the value of $F_{thres}$; a fourth column that includes the number of additional routes included when locating the sensors in the best set obtained for each case; and a last column that considers all the routes in C used in the estimation model. In this table, it can be seen that, with less simplification, the set of routes in C with which we work is greater, and therefore the estimations are better. As the simplification increases, more routes are simplified and this means that, by locating the sensors, a greater number of routes are recovered, but a set C on similar order of magnitude.

Finally, Table 6 indicates the SL sets obtained for the cases where $F_{thres}$ is equal to 10 and 15, since they offer the best results and where there is little difference in the best RMARE estimation error. We see how for both sets, there is only a difference of 8 links from the 30 considered, the rest being common in both. For both sets, constraint (6) is considered in the location model, and therefore no links belonging to NSL appears. Furthermore, this tells us how, depending on how the network is modeled, one set or another may be obtained, with small differences but which may influence the observability and estimation results of the network.

## 4. Conclusions

This paper presents a proposal for deployment of a low-cost sensor network for automated vehicles plate recognition in a pilot project in Ciudad Real (Spain). For this, three main tools were needed: (1) the architecture to deploy the sensors, (2) a low-cost sensor prototype, and (3) a methodology to decide the best location for the sensor.

Regarding the deployment of sensors and the sensors themself, one of main features to highlight is that the total cost is very low in terms of the following elements:Production/Manufacturing: The unit cost of the hardware components for the realization of the prototype is less than €60 (considering the tripod as an extra accessory). In the case of integration for large scale manufacturing, these could be significantly reduced.Installation: The sensors have a very low energy consumption, which allows their deployment in any location and without specific energy supply infrastructure. The platform allows adapting the sensor parameters (resolution, lighting levels, shutter speed, and compression level) to the specific needs of each location.Maintainability and scalability: The proposed architecture allows working with any existing ANPR library in the market by delegating tasks between processing layers, as well as their combination to improve the overall success rate. The detection stage is delegated to the smart management layer, reducing overall costs, and providing more scalable and efficient solutions.

In addition, the deployed sensor is completely decoupled from the specific license plate identification platform used. This allows to change the platform if the user found any other better. In particular, the used platform identifies besides the license plate number, the vehicle’s manufacturer, model, and color data. This information can be used in the overall analysis of traffic flows with a view to reducing possible errors in the identification of the number plate and will be developed in the future by the authors.

The third tool used in this paper is a methodology to determine the location in the traffic network of the designed sensors. To this, we have proposed the use of two algorithms which aim to achieve a good enough quality of the traffic flow estimation to be done (in terms of low RMARE value) with the ANPR data collected by the sensors.

The model was applied to the traffic network of a pilot project considering a deployment of 30 sensors analyzing whether or not to install the proposed sensors on some links due to the difficulty of its installation. The results were very positive since the expected quality of the estimation results is very similar to those obtained when allowing the sensor to be located in any link. The main advantage is that avoiding those conflictive links we expect a reduction obtaining errors of reading vehicle plates.

The influence of other parameters of the model were also analyzed such as the number of routes used as reference and the degree of network simplification. The analysis of the results shows that considering a greater number of reference routes, represented by means of the parameter k, leads to a better estimation of the flows in terms of achieving a smaller RMARE. However, a high value for k would imply working with a network with a large number of routes, which would have a high computational cost. In reference to network simplification, a medium–low degree of network simplification leads to a good performance of the methodology in terms of the error obtained in the estimation step.

## Figures and Tables

**Figure 1 sensors-20-05589-f001:**
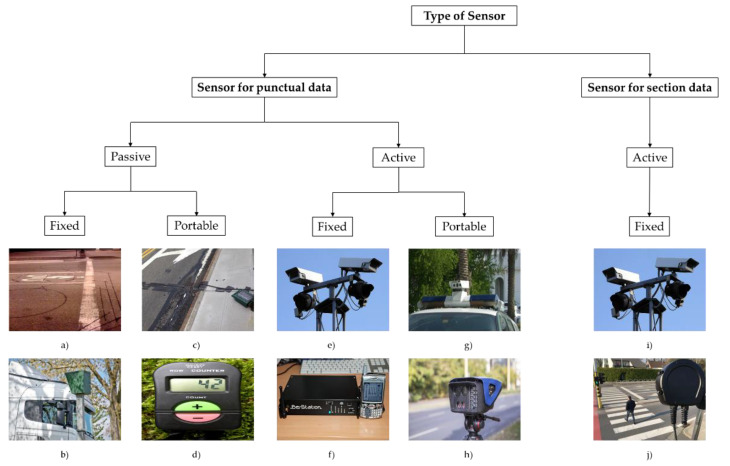
Classification and examples of sensors according to the characteristics of the collected data, their interaction with the vehicle and position on the road: (**a**) Inductive loop detector; (**b**) Microwave radar; (**c**) Rubber hoses detector; (**d**) Hand electronic counter; (**e**) and (**i**) Automatic Number Plate Recognition (ANPR) fixed sensors; (**f**) Bluetooth sniffer; (**g**) Police ANPR sensor on vehicle; (**h**) Police ANPR portable sensor; (**j**) Bluetooth scanning sensor.

**Figure 2 sensors-20-05589-f002:**
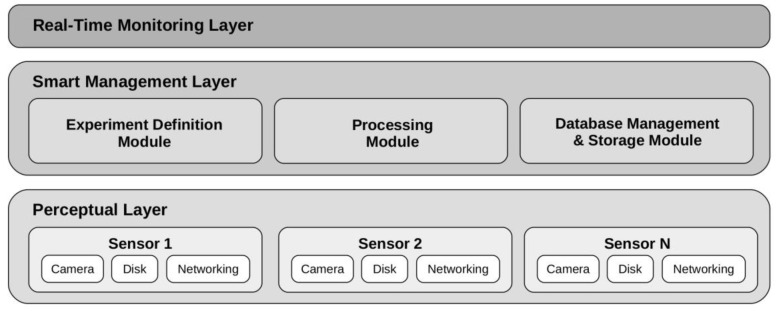
Multi-layer architecture for the deployment of low-cost sensor networks for automatic license plate recognition.

**Figure 3 sensors-20-05589-f003:**
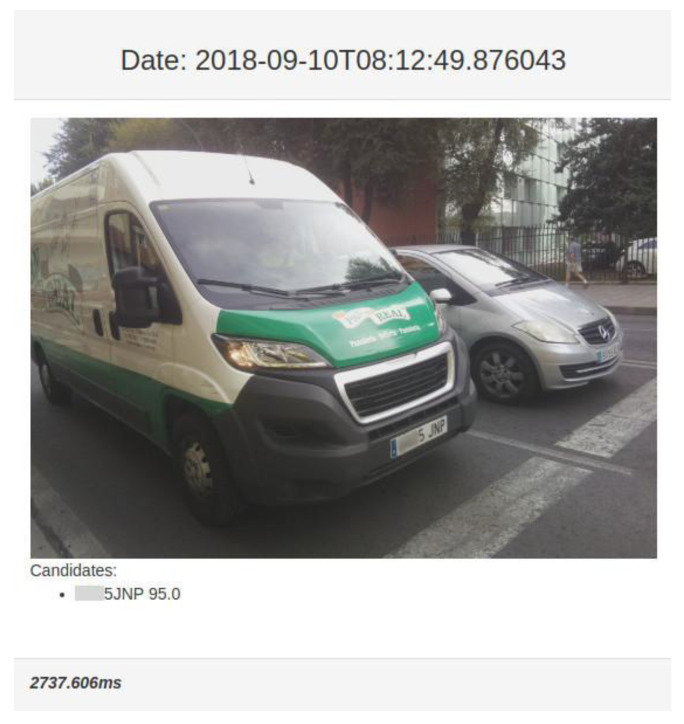
Visualization obtained from one of the sensors. (To protect personal data, the first three digits of the license plate have been blurred.)

**Figure 4 sensors-20-05589-f004:**
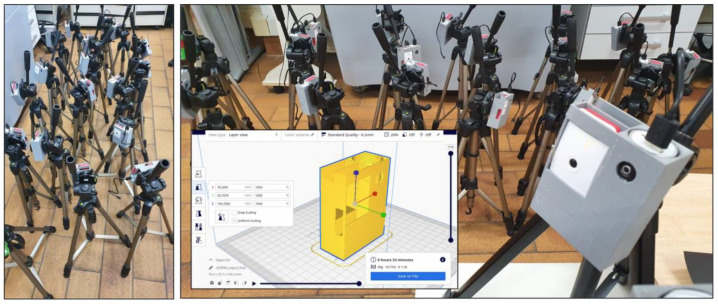
Prototype of the designed sensor.

**Figure 5 sensors-20-05589-f005:**
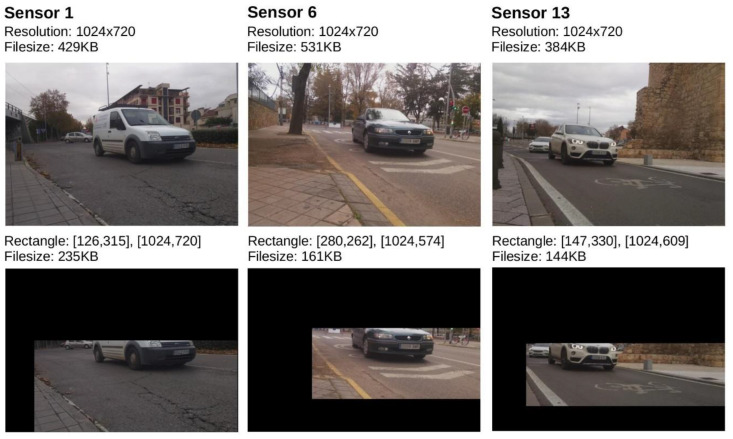
Example of definition of clipping parameters in capture sub-regions in three sensors of the deployed system, with comparative analysis of storage size for each frame. (To protect personal data, the first three digits of the license plate have been blurred).

**Figure 6 sensors-20-05589-f006:**
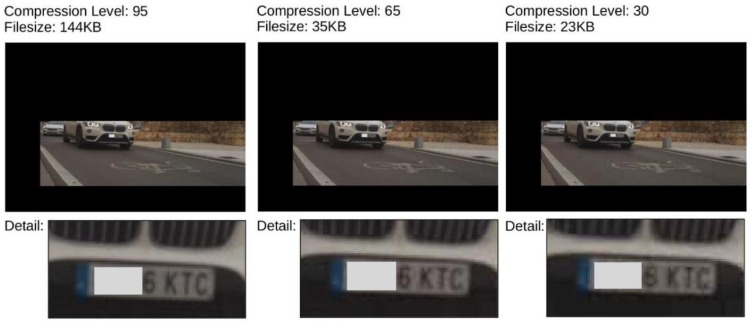
Different compression levels of the JPG standard. With values below 60, with significant loss of high frequency information, the image quality significantly compromises the success rate of the license plate detection algorithms. (To protect personal data, the first three digits of the license plate have been blurred).

**Figure 7 sensors-20-05589-f007:**
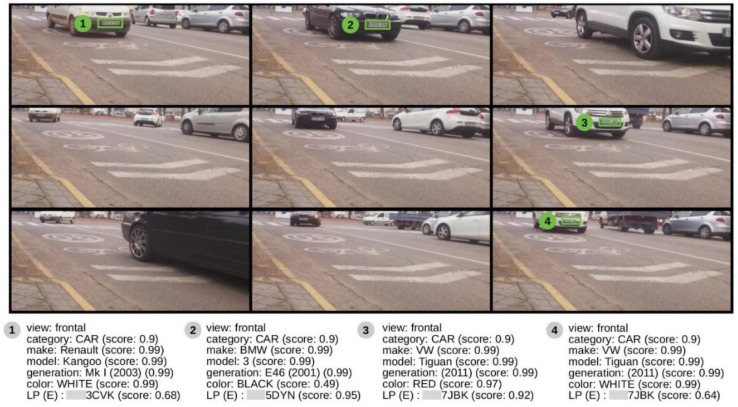
Detection of vehicles in a 3 × 3 image matrix, which allows, by means of a single sending to the cloud, to summarize 9 s of traffic analysis of a given sensor. The system detects both the number plate and certain characteristics of the vehicle, assigning a confidence value to each detection. (To protect personal data, the first three digits of the license plate have been blurred).

**Figure 8 sensors-20-05589-f008:**
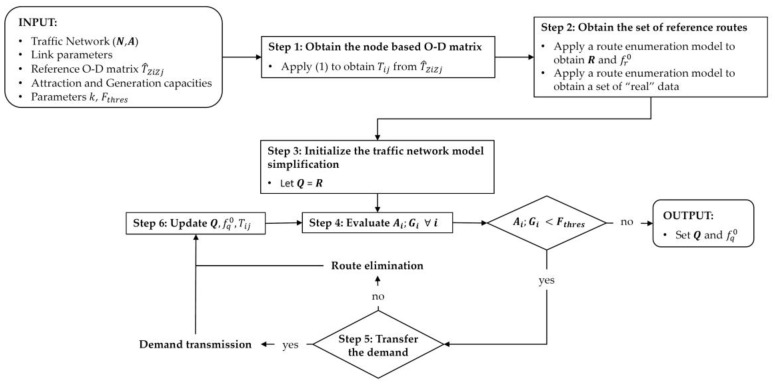
Flowchart of the algorithm defined for the traffic network modelling.

**Figure 9 sensors-20-05589-f009:**
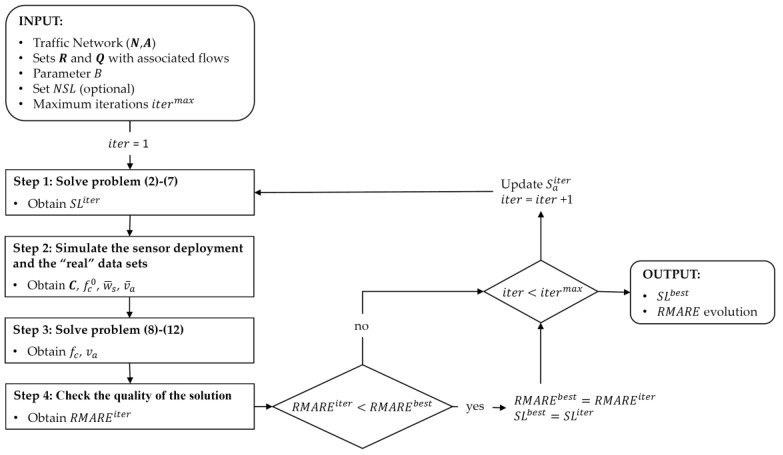
Flowchart of the algorithm defined for the ANPR sensor location model.

**Figure 10 sensors-20-05589-f010:**
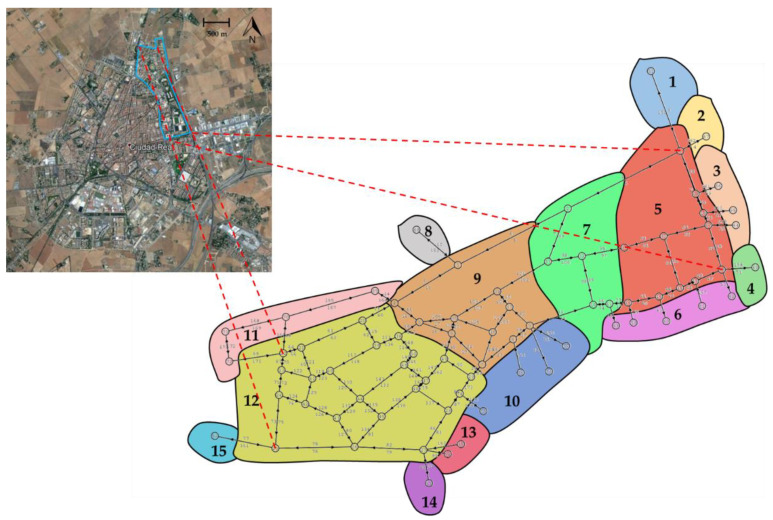
Traffic network to be modelled for analysis: Plan view of the urban area of Ciudad Real and the delimited Campus area to be modelled; Ciudad Real Campus network and its division in 15 traffic zones.

**Figure 11 sensors-20-05589-f011:**
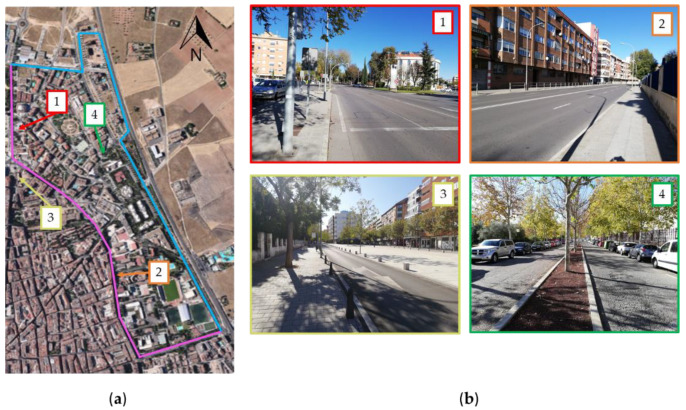
(**a**) Representation of the corridor, in violet, where the greatest flows and difficulties in installing a sensor take place; (**b**) Images of several links on the network.

**Figure 12 sensors-20-05589-f012:**
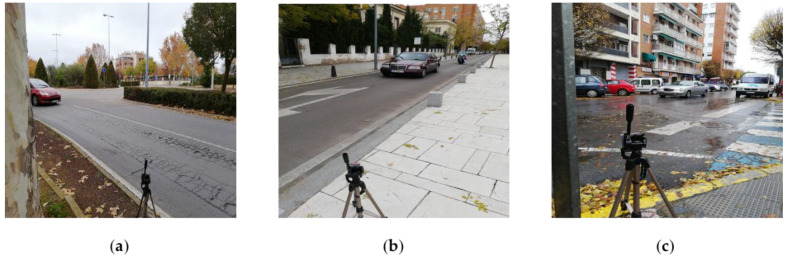
Examples of installed sensor at strategic points: (**a**) Exit from an intersection and enter a link with two lanes; (**b**) Link with a single lane with unidirectional flow; (**c**) Entrance to a link with a single lane.

**Figure 13 sensors-20-05589-f013:**
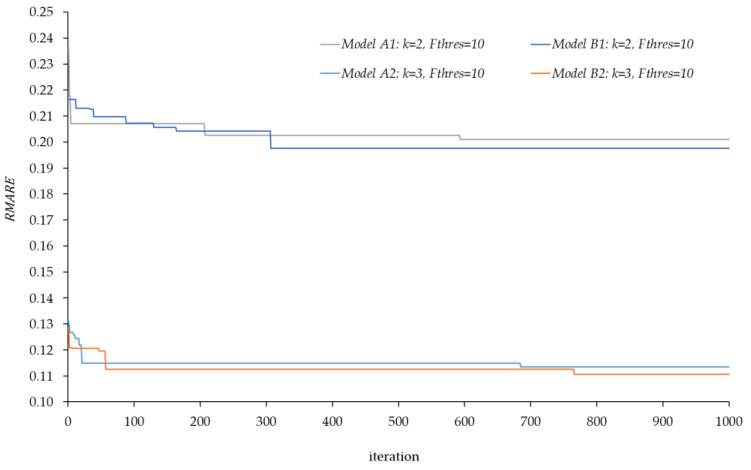
Evolution of RMARE value for the different cases varying the k parameter and the threshold flow.

**Figure 14 sensors-20-05589-f014:**
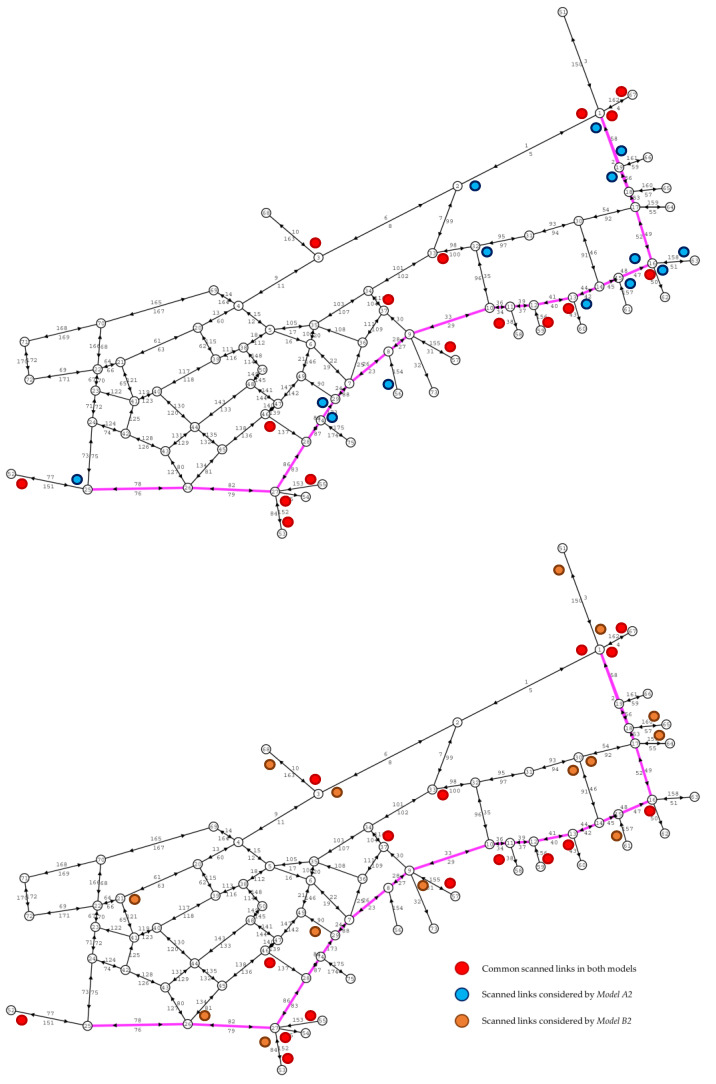
Upper figure: Optimal sensor locations considering k = 3, $F_{thres}$ = 10, and all links as candidates to be scanned; Lower figure: Optimal sensor locations considering k = 3, $F_{thres}$ = 10, and a certain set non-candidate scanned links (NSL) of links as no candidates to be scanned.

**Figure 15 sensors-20-05589-f015:**
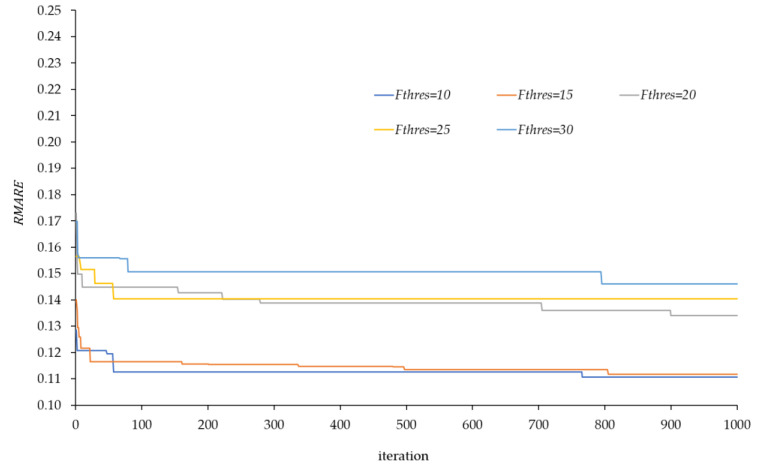
Evolution of RMARE value for the different cases varying the threshold flow.

**Table 1 sensors-20-05589-t001:** Power consumption comparison in mAh of different versions of Raspberry Pi.

	Zero	ZeroW	A	B	Pi2B	Pi3B
Idle	100	120	140	360	230	230
Capture	140	200	260	420	290	290
Networking	-	230	320	480	350	350

**Table 2 sensors-20-05589-t002:** List of nodes on the traffic network and their associated zones.

Node	Zone	Node	Zone	Node	Zone	Node	Zone	Node	Zone	Node	Zone
1	5	14	5	27	12	40	12	53	14	66	3
2	7	15	5	28	12	41	12	54	13	67	2
3	9	16	5	29	9	42	12	55	13	68	8
4	9	17	5	30	5	43	12	56	10	69	11
5	9	18	5	31	5	44	12	57	10	70	11
6	9	19	5	32	7	45	12	58	6	71	11
7	9	20	12	33	7	46	12	59	6	72	11
8	9	21	12	34	9	47	12	60	6	73	10
9	9	22	12	35	9	48	12	61	6	74	12
10	7	23	12	36	9	49	9	62	6	75	10
11	5	24	12	37	9	50	12	63	4		
12	5	25	12	38	12	51	1	64	3		
13	5	26	12	39	12	52	15	65	3		

**Table 3 sensors-20-05589-t003:** O–D trip matrix per defined zones.

Zone	1	2	3	4	5	6	7	8	9	10	11	12	13	14	15	Total
1	-	269	0	31	21	115	17	0	68	0	10	23	60	217	137	968
2	125	-	0	83	13	97	0	0	0	25	0	45	23	36	8	455
3	0	0	-	170	0	64	0	17	0	35	0	0	0	28	0	314
4	124	160	0	-	156	83	118	0	0	0	46	0	166	224	0	1077
5	70	89	0	0	-	0	0	0	0	0	0	118	0	40	5	322
6	43	118	0	70	20	-	46	32	0	0	2	0	0	48	28	407
7	50	12	0	0	0	0	-	0	0	0	0	9	0	17	2	90
8	3	4	0	0	0	31	0	-	0	47	12	34	25	16	0	172
9	15	26	48	62	0	55	0	0	-	105	39	73	64	13	22	522
10	30	12	0	55	0	37	0	95	375	-	0	0	57	21	0	682
11	8	8	0	0	0	0	0	0	0	0	-	26	0	37	3	82
12	245	29	0	0	0	154	0	44	0	0	0	-	0	0	28	500
13	105	39	0	181	0	106	0	0	0	0	0	0	-	63	0	494
14	0	83	0	271	98	461	75	37	26	83	50	0	0	-	0	1184
15	80	25	0	0	123	144	31	0	52	10	21	0	0	0	-	486
Total	898	874	48	923	431	1347	287	225	521	305	180	328	395	760	233	7755

**Table 4 sensors-20-05589-t004:** Best set of links (SL) sets obtained from the variability analysis of k parameter.

Model	Scanned Link Set SL	RMARE
*A1*	**1** 2 **3** 5 **10** 27 **31** 42 **43** 47 48 **50** 51 56 58 60 **85** 89 100 110 **137 150 151 152 153 154 155 156** 158 **163**	0.2010
*B1*	**1 3** 4 **10** 20 **31** 38 **43 50** 54 63 64 77 84 **85** 90 94 **137 150 151 152 153 154 155 156** 157 159 161 162 **163**	0.1976
*A2*	**1** 2 **4** 5 **10 38** 42 **43** 47 48 **50** 51 56 58 77 **85** 89 97 **100 110 137 151 152 153** 154 **155 156** 158 **162** 173	0.1149
*B2*	**1** 3 **4** 8 **10** 31 **38 43 50** 63 81 84 **85** 90 91 92 **100 110 137** 150 **151 152 153 155 156** 157 159 160 **162** 163	0.1106

**Table 5 sensors-20-05589-t005:** Number of routes that appear according to the considered $F_{thres}$ value.

$F_{thres}$	Number of Routes in R	Number of Routes in Q	Added Routes Compatibles with SL	Number of Routes in C
10	2943	2896	23	2919
15	2943	2816	30	2846
20	2943	2398	31	2429
25	2943	2175	41	2216
30	2943	2090	47	2137

**Table 6 sensors-20-05589-t006:** Best SL sets obtained from the variability analysis of k parameter.

$F_{thres}$	Scanned Link Set SL	RMARE
10	1 **3** **4** 8 10 **31** **38** **43** **50** 63 81 **84** **85** **90** **91** **92** 100 110 **137** **150** **151** **152** **153** **155** **156** **157** 159 **160** **162** **163**	0.1106
15	**3****4** 6 7 19 25 **31** **38** **43** **50** 60 72 **84** **85** **90** **91** **92** **137** **150** **151** **152** **153** 154 **155** **156** **157** **160** 161 **162** **163**	0.1117

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
