# Peer review of "A Low-Cost Automatic Vehicle Identification Sensor for Traffic Networks Analysis"

_sensors, 2020, doi:10.3390/s20195589_

Round 1
Reviewer 1 Report
I do not have new comments.
My previous comments receive the demanded improvements.
Author Response
Comment #1: I do not have new comments. My previous comments receive the demanded improvements.
Answer: Thank you for your previous report and for the time taken reading our paper.
Reviewer 2 Report
The authors have addressed most of my concerns. In terms of state-of-the-art literature about so-called vehicular sensing networks, some important theoretical works are missing, such as ' Vehicular sensing networks in a smart city: Principles, technologies and applications', ' Internet of vehicles: Sensing-aided transportation information collection and diffusion', just to name a few.
Author Response
Dear Reviewer, Once more, we want to thank you for the time taken in this review. Next, we present the answers to your concern, and in the new version of the manuscript and highlighted in green, the changes made in the paper due to it.
Comment #1: The authors have addressed most of my concerns. In terms of state-of-the-art literature about so-called vehicular sensing networks, some important theoretical works are missing, such as ' Vehicular sensing networks in a smart city: Principles, technologies and applications', ' Internet of vehicles: Sensing-aided transportation information collection and diffusion', just to name a few.
Answer: We have modified a paragraph of the introduction to be able to make a first clearer differentiation or classification of the types of sensors used for traffic analysis. With this first classification, we have considered introducing these works that we think enrich the article and the consulted references. Apart from other minor changes (see lines 46 and 83), the following text have been included form line 38 to line 46:
In particular, [1] differs between in-vehicle and in-road sensors. The first are those that allow increasing the performance of the driving and the connectivity of the vehicle with their environment. In this, the concepts of communication between vehicles and the Vehicular Sensing Networks (VSN) are called to be important in the improvement of the quality and operability of transportation systems (see [2] and [3]). The second are those installed in the transportation network and allows to monitor the performance of the system to, according to the extracted information, diagnose the problems, improve the resilient and operational functioning, and inform the users helping them to make better choices. In this paper we mainly focus in this last
References:
|
[1] |
J. Guerrero-Ibañez, S. Zeadally and J. Contreras-Castillo, "Sensors Technologies for Intelligent Transportation System," Sensors, vol. 18, no. 2, pp. 1-24, 2018. |
|
[2] |
J. Wang, C. Jiang, Z. Han, Y. Ren and L. Hanzo, "Internet of Vehicles: Sensing-Aided Transportation Information Collection and Diffusion," IEEE Transactions on Vehicular Technology, vol. 67, no. 5, pp. 3813-3825, 2018. |
|
[3] |
J. Wang, C. Jiang, K. Zhang, T. Quek, Y. Ren and L. Hanzo, "Vehicular Sensing Networks in a Smart City: Principles, Technologies and Applications," IEEE Wireless Communications, vol. 25, no. 1, pp. 122-132, 2018. |
Reviewer 3 Report
In this manuscript, it is exposed the deployment of a network of image sensors (cameras) for plate numbers recording, in order to estimate the traffic flow in Ciudad Real (Spain). The manuscript explains how the low-cost image sensors were built, how the network and software infrastructure is design, and how the optimization algorithm defines the placement of the image sensors. The evaluation of the proposed methodology was tested on a real scenario.
It is my opinion that the main contributions of this manuscript are 1) the analysis of proposed methodology using real data, and 2) the integration of different technologies to build a low-cost image sensor whose cost is around $ 60. I understand that the optimization algorithm for sensor placement has been previously published by the same authors (Ref. [10]).
In general, the paper is clear and scientifically sound. Nevertheless, the following aspects should be addressed before considering for its publication.
1) In figure 3, it is shown the number plate of a vehicle with a complete date. I think that it must be clarified if this photograph does not violate any privacy policy. This applies to any photograph in the paper that exposes a plate number and complete date.
2) Line 410. "The image has a size of 432MB" may be confusing since the paper previously talks about details of the Sony image sensor. I recommend using "installation image" instead.
3) Any discussion in this paper that is also exposed in Ref. [10] has to be eliminated for the sake of brevity.
Although I am not a native English speaker, I may have found the following grammatical errors in the article:
1) Line 31, "...is a problem that has been tackling now for decades", could be "has been tackled". Please, check.
2) Line 166. "(under 25 mph)". I think that Sensors journal is more comfortable with SI units, so maybe km/h should be used instead of mph. Please, check.
3) Line 179, " in intel processor" is "Intel" with uppercase.
4) Line 216, "in Section 2, the proposed low-cost sensor and its associated system for traffic networks analysis is deeply described" is "... are deeply described". Please, check.
5) Line 254, "... is done in section." What section?. Please, check.
6) Line 371. "As previously stated in section X.Y". Please, indicate the correct section number.
Author Response
Comment #1: In this manuscript, it is exposed the deployment of a network of image sensors (cameras) for plate numbers recording, in order to estimate the traffic flow in Ciudad Real (Spain). The manuscript explains how the low-cost image sensors were built, how the network and software infrastructure is design, and how the optimization algorithm defines the placement of the image sensors. The evaluation of the proposed methodology was tested on a real scenario.
Answer: This is a good summary of the contributions of the paper. Next, we present the answers to the reviewer’s concerns, hoping that they match the expected. In the new version of the manuscript (and highlighted in blue) the reviewer can find the changes made in the paper due to his/her comments.
Comment #2: It is my opinion that the main contributions of this manuscript are 1) the analysis of proposed methodology using real data, and 2) the integration of different technologies to build a low-cost image sensor whose cost is around $ 60. I understand that the optimization algorithm for sensor placement has been previously published by the same authors (Ref. [10]).
Answer: With due respect, as Section 1.3 points out, the paper proposes a third main contribution which is a methodology to locate sensors (not just an optimization algorithm). It is true that the methodology is based on the work done in a previous work (Ref. [10]), but the step forward given in this paper is that we have modified the model to take into account the special characteristics of the sensor installation (see section 2.3). Section 3.1. deals with this problem for the particular case or our pilot project network. Furthermore, using the proposed methodology, we have proved that the expected quality of the traffic flow estimation results are very similar if the sensor can be located in any link compared with avoiding links with certain problems to install the sensor.
In order to better highlight this third contribution, we have added this last comment in line 222.
Comment #3: In general, the paper is clear and scientifically sound. Nevertheless, the following aspects should be addressed before considering for its publication.
Comment #3.1. In figure 3, it is shown the number plate of a vehicle with a complete date. I think that it must be clarified if this photograph does not violate any privacy policy. This applies to any photograph in the paper that exposes a plate number and complete date.
Answer: Although there is an understanding about the fact that number plates are not considered, in Spain, as personal data, we thought it best to blur the ones that are clearly visible in the paper. This has been done in Figure 3, Figure 5, and Figure 6 and Figure 7.
Comment #3.2. Line 410. "The image has a size of 432MB" may be confusing since the paper previously talks about details of the Sony image sensor. I recommend using "installation image" instead.
Answer: The term “installation image” has been used in line 424 (old 410).
Comment #3.3. Any discussion in this paper that is also exposed in Ref. [10] has to be eliminated for the sake of brevity.
Answer: We really believe that a paper has to be self-understood (i.e. without the need of reading other related papers unless the reader would like to delve into the matter). This was the reason why we include a complete description of the model exposed in Ref [10]. In addition, to understand the step forward mentioned in Comment#2, the paper needs the analysis already done of the method. Therefore, and with due respect, we prefer to keep the section as it is, because by shortening the paper we could incur in a loss of clarity.
Comment #4. Although I am not a native English speaker, I may have found the following grammatical errors in the article:
Comment #4.1. Line 31, "...is a problem that has been tackling now for decades", could be "has been tackled". Please, check.
Answer: Line 31 has changed to “is a problem that has been tackled for decades”.
Comment #4.2. Line 166. "(under 25 mph)". I think that Sensors journal is more comfortable with SI units, so maybe km/h should be used instead of mph. Please, check.
Answer: This has been modified in lines 175 and 176.
Comment #4.3. Line 179, " in intel processor" is "Intel" with uppercase.
Answer: “intel” has been replaced by “Intel” in current line 188.
Comment #4.4. Line 216, "in Section 2, the proposed low-cost sensor and its associated system for traffic networks analysis is deeply described" is "... are deeply described". Please, check.
Answer: Current line 228 now indicates that “in Section 2, the proposed low-cost sensor and its associated system for traffic networks analysis are deeply described”
Comment #4.5. Line 254, "... is done in section." What section?. Please, check.
Answer: Line 267 now shows that “... is done in section 2.2”.
Comment #4.6. Line 371. "As previously stated in section X.Y". Please, indicate the correct section number.
Answer: Line 386 now shows that “.As previously stated in section 2.1.3”.
This manuscript is a resubmission of an earlier submission. The following is a list of the peer review reports and author responses from that submission.
Round 1
Reviewer 1 Report
This manuscript proposes a three-layer architecture to deploy low-cost sensor networks for automatic license plate detection to address the problem of high manufacturing and installation cost. A case of study with the installation of proposed devices is presented to demonstrate its viability.
Strong Points:
- The topic is meaningful and the proposed architecture is feasible.
- A comprehensive case study is made to verify the feasibility of the proposed architecture and the corresponding system is in use in practical situations.
Weak Points:
- Scientific contributions are limited. There are not any novel algorithms or novel primitive models proposed.
- No comparisons are made. As is indicated in the manuscript, most existing literature do not focus on the theme of the cost, and how to draw this conclusion? What is the advantage of the proposed architecture against other architectures?
- Real or simulated data are in need to demonstrate the percentage of reduced cost of the proposed architecture compared to common settings.
Reviewer 2 Report
The pilot project is relevant for the research.
The structure deviates from the usual. The general impression: many information are thrown with lack of organization. The article structure has to be improved.
“Introduction” section contains beside the presentation of the solved problems many parts usual included into “Related works” and “State of the art”. I prefer the classical structure. It is easier the understanding of the authors’ contribution relative to the previous existing solutions.
Section 2 has a too long title. Beside the architecture, there are included many information concerning the processing cost mixing hardware details with software details. Mixing the hardware and software information with the processing details is confusing the reader. I suggest to split them.
This has to be done for the entire article.
The term “real-time” is used careless. Its origin is from real-time systems characterizing application with temporal constraints. You have used often for on-line systems without providing any information if the responses (reactions) met heir deadlines. This has to be clarified and correctly analyzed.
Related to the network modeling: is confusing (again!) mixing the structure with the behavior. This has to be clarified also.
Subsection “Sensor location model”: is mixing the model with the search of solution. A clear classical algorithm with inputs, outputs and the processing parts would help the understanding.
A general observation: a formal approach would help the understanding of your achievements.